# Impact of Central Sensitization on Clinical and Functional Aspects of Psoriatic Arthritis

**DOI:** 10.3390/medicina60091449

**Published:** 2024-09-04

**Authors:** Mehmet Nur Kaya, Duygu Tecer, Özlem Kılıç, Merve Sungur Özgünen, Sedat Yılmaz

**Affiliations:** Rheumatology Department, Gülhane Training and Research Hospital, University of Health Sciences Turkey, 38000 Ankara, Turkey; duygu-tecer@hotmail.com (D.T.); ozlemk.kara@gmail.com (Ö.K.); merve.sungur.dr@gmail.com (M.S.Ö.); drsy75@gmail.com (S.Y.)

**Keywords:** psoriatic arthritis, central sensitization, Central Sensitization Inventory

## Abstract

*Background/Objectives*: Psoriatic arthritis (PsA) is an inflammatory rheumatic disease characterized by peripheral arthritis, enthesitis, spondylitis and psoriasis. The objective of this study was to examine the prevalence of central sensitization (CS) and its impact on the clinical and functional aspects of PsA. *Methods*: Adult patients with PsA according to the Classification of Psoriatic Arthritis (CASPAR) criteria were included in this cross-sectional observational study. The Central Sensitization Inventory (CSI) was used to assess the presence of CS. The study evaluated the impact of CS on individuals by analyzing many factors including demographic information, laboratory findings, clinical features, disease activity, quality of life, severity of sleeplessness, frequency of depression and anxiety. The patients were categorized into distinct groups based on the existence and intensity of CS, and a comparative analysis was conducted on their respective outcomes. *Results*: A total of 103 PsA patients with a mean age of 43.2 (SD: 6.7) years and including 42 (40.8%) males were included. The mean CSI score was 45.4 (SD: 15.1), and 67 (65.1%) patients had CS. The logistic regression analysis revealed that the variables Psoriasis Area Severity Index (PASI), General Anxiety Disorder-7 (GAD-7), and Insomnia Severity Index (ISI) exhibit considerable predictive power in relation to the outcome variable CS (*p* < 0.05). PASI was observed as the most important variable in predicting CS (OR 9.70 95% CI: 1.52–62.21). *Conclusions*: CS has demonstrable efficacy in influencing laboratory, clinical, and functional markers among individuals with PsA. When assessing pain sensitivity in these patients, it is important to take into account the presence of CS.

## 1. Introduction

Psoriatic arthritis (PsA) is a heterogeneous chronic immune-mediated inflammatory disease associated with psoriasis. The affected organ systems encompass a wide range of areas, including peripheral and axial joints, entheses, skin and nails [1]. The primary aim of therapeutic interventions is optimizing functional status, improving quality of life, and preventing structural damage through the controlling symptoms and inflammation. A recommended treatment target is achieving remission or low disease activity [2]. However, pain may persist even in PsA patients who achieve minimal disease activity and is associated with deterioration in quality of life, a decrease in physical functionality, limitation in activities of daily living, and a decrease in work productivity [3,4,5].

Pain is one of the variables included in the main composite disease activity measurement indices for PsA. These indices represent a facilitating tool for shared decision making in daily clinical practice and an efficient assessment tool in observational studies and clinical trials [6]. Therefore, accurate assessment of pain is essential, because although the pain associated with active inflammatory joint disease is categorized as nociceptive pain, patients may also experience neuropathic, nociplastic, or mixed pain [7,8]. The term nociplastic pain was first proposed in 2016 as a third mechanistic descriptor for chronic pain states, mechanistically different from the nociceptive and neuropathic pain [9]. Nociplastic pain is defined by the International Association for the Study of Pain (IASP) as pain that arises from altered nociception despite no clear evidence of actual or threatened tissue damage causing the activation of peripheral nociceptors or evidence for disease or lesion of the somatosensory system causing the pain [10]. It is characterized by hyperalgesia (increased pain in response to noxious stimuli) and allodynia (pain as a response to normally non-painful stimuli) [11]. Central sensitization (CS), a key underlying mechanism of nociplastic pain, is defined as the elevated responsiveness of nociceptive neurons in the central pain pathway to normal or subthreshold afferent inputs as a result of central nervous system (CNS) plasticity [12]. Although assessments of CS in humans remain challenging, quantitative sensory testing (QST) is one of the most commonly reported methods of measuring altered somatosensory function. QST provides indirect evidence of central sensitivity including reduced pain thresholds, greater pain in response to suprathreshold stimulation, and temporal summation [13]. The Central Sensitization Inventory (CSI) is a patient questionnaire that primarily assesses symptoms considered to be related to CS. CSI is a reliable, consistent, and valid instrument for quantifying the severity of several symptoms of CS [14].

In patients with rheumatic disease, ongoing nociceptive stimulus from a joint can cause neuroplastic changes in the CNS [8]. Currently, there are few data on the prevalence of CS in patients with PsA. In the first study assessing CS using the CSI questionnaire in PsA patients, CS was detected in 42.9% of patients and independently associated with functional disability in PsA. Furthermore, it is currently unknown to what extent CS interferes with the assessment of clinical disease activity or affects the functional ability of such individuals [15].

However, in the aforementioned study, since patients diagnosed with fibromyalgia (FM) and/or major depressive disorder were excluded, their findings cannot be generalized to the overall PsA population. Therefore, the objective of this study was to examine the phenomenon of CS in individuals diagnosed with PsA and to explore the potential relationship between CS, its severity, disease activity, anxiety, depression, sleep quality, and functionality.

## 2. Materials and Methods

The study was carried out with the permission of Ethics Committee of Health Sciences University, Gülhane Training and Research Hospital (Date: 12 April 2023, Decision No: 2023/70). Well-written informed consent was obtained from all participants according to the principles of the Helsinki Declaration.

### 2.1. Study Design and Sampling

This study was planned as a cross-sectional and single-center study. Between April 2023 and July 2023, consecutive PsA patients admitted to Gülhane Training and Research Hospital Rheumatology outpatient clinics were evaluated for inclusion. All individuals enrolled in this research were diagnosed with psoriasis (PsO). Patients aged 18–75 years and who met the International Classification of Psoriatic Arthritis (CASPAR) classification criteria were included. Patients were excluded if they had met any of the following criteria: previously established diagnosis of psychiatric disease (e.g., major depressive disorders, generalized anxiety disorder), receiving centrally acting drugs (amitriptyline, duloxetine, pregabalin, opioids, corticosteroids), concomitant inflammatory rheumatic disease, comorbid chronic diseases (e.g., hyperthyroidism, hypothyroidism, diabetes mellitus, and malignancies), age < 18 years, unable to understand and complete questionnaires, or inability to give written informed consent.

### 2.2. Clinical Variables

Demographic variables including age, gender, educational status, current smoking status, body mass index (BMI) and clinical variables including age at diagnosis, time since diagnosis, family history of PsO, HLA-B27 status, presence of peripheral arthritis, history of inflammatory bowel disease (IBD) and uveitis, entheseal and sacroiliac joint involvement, and drug use (NSAIDs, methotrexate and biologic agents) were recorded.

The severity of psoriasis was assessed by calculating Psoriasis Area and Severity Index (PASI). The size of the affected area is expressed as a percentage, and its severity (desquamation, induration, erythema) as points. Disease activity is evaluated over the score obtained by combining the size and severity of the affected area [16].

C-reactive protein (CRP) and erythrocyte sedimentation rate (ESR), Disease Activity in Psoriatic Arthritis (DAPSA), and Disease Activity Score-28 (DAS-28) were used to assess the disease activity of PsA. DAS28 scoring is as follows; >5.1 indicates high disease activity, 3.2–5.1 moderate disease activity, 2.6–3.2 low disease activity, and <2.6 remission [17]. DAPSA scores are as follows: 0–4 indicates remission, 5–14 low disease activity, 15–29 moderate disease activity, and 28 high disease activity [18].

### 2.3. Outcome Measures

The Central Sensitivity Inventory consists of two parts. Part A consists of a 25-item questionnaire used to assess CS-related symptoms, with each item is scored on a 5-point Likert scale ranging from 0 = “never” to 4 = “always”. The total score ranges from 0 to 100. Classification according to severity level is as follows; 0–29: subclinical; 30–39: mild; 40–49: moderate; 50–59: severe; and 60–100: excessive [19]. A score ≥ 40 indicates a high likelihood of CS [20]. Part B investigates the presence of 10 central sensitivity syndromes. The CSI questionnaire utilized in our study was the Turkish version, which has been validated by Düzce E et al. [21].

General Anxiety Disorder-7 (GAD-7) is a well-established, satisfactorily reliable and valid scale that is widely used to evaluate anxiety. It consists of 7 items and is scored over a total of 21 points. The results are evaluated as follows: 0–5 indicates minimal anxiety, 6–10 mild anxiety, 11–15 moderate anxiety, and 16–21 severe anxiety [22]. The Turkish version of the GAD-7 questionnaire was used in this study [23].

The Insomnia Severity Index (ISI) is a well-recognized index with confirmed reliability, sensitivity, and validity. It consists of 7 items and is scored over a total of 28 points. The total score is used to determine the degree of insomnia. Total scores are evaluated as follows: 0–7 indicates no clinically significant insomnia, 8–14 subthreshold insomnia, 15–21 moderate clinical insomnia, and 22–28 severe clinical insomnia [24]. The Turkish valid version of the ISI questionnaire was made by Boysan et al. [25].

The Patient Health Questionnaire-9 (PHQ-9) is a 9-item scale scored over 27 points that measures depression. It is a well-established and reliable scale that is used to evaluate depression. The results are evaluated as follows: 0–4 indicates no depression, 5–9 mild depression, 10–14 moderate depression, 15–19 moderately severe depression, and 20–27 severe depression [26]. A validation process of the Turkish version of the PHQ-9 questionnaire was conducted [27].

The Health Assessment Questionnaire (HAQ) is a scale consisting of 8 sections and 20 items, evaluating activities of daily living. Each item is scored between 0 and 3 (0 = without any difficulty; 1 = with some difficulty; 2 = with much difficulty; and 3 = unable to do). The sections that make up the scale are dressing, straightening, eating, walking, hygiene, reaching, grasping and daily tasks, and each section contains two or three items. Each section is scored separately, and a single HAQ score, which can vary between 0 and 3, is determined by taking the average of the scores of the 8 sections [28]. A validation process of the HAQ questionnaire in the Turkish language was conducted [29].

### 2.4. Statistical Analysis

The statistical analyses were conducted using version 28 of Statistical Package for Social Sciences (Armonk, NY, USA: IBM Corp) program for Windows. In the power analysis, the sample size was taken as effect size 0.5 and was determined to include at least 106 patients. The Shapiro–Wilk test was used to evaluate whether the variables were normally distributed. Normally distributed variables were expressed as mean ± standard deviation (SD), skewed variables as median (interquartile range-IQR), and categorical variables as number and percentage (%).

The study population was divided into two groups using a CSI cutoff score ≥ 40, which indicated high probability of having CS. Patients’ demographic and clinical characteristics were compared between these two groups using the chi-square test or Fisher’s exact test (when chi-square test assumptions do not hold) for categorical data, the Mann–Whitney U-test for non-normally distributed continuous data, and an independent T-test for normally distributed continuous data. In addition, the study population was divided into five groups according to their CS severity level. Clinical characteristics among the CS severity level were compared using a one-way ANOVA for parametric variables and the Kruskal–Wallis test for non-parametric variables. When overall significance was observed, pairwise post hoc tests were performed using Tukey’s test or the Mann–Whitney U-test with Bonferroni correction, where appropriate. The correlation between CSI score and clinical variables was assessed using the Pearson test for variables.

A forward stepwise multivariate regression model was developed, using factors that had significant associations with CS in the univariate analysis. Statistical significance was determined by accepting *p*-values that were less than 0.05.

## 3. Results

A total of 103 PsA patients with a mean age of 43.2 (SD: 6.7) years and consisting of predominantly females (59.2%) were included in this study. Although all of the patients included in the study had PsO, 31 (30%) patients had a family history of PsO, 4 (3.8%) patients had IBD, 21 (20.3%) patients had a previous occurrence of uveitis, 56 (54.3%) patients had entheseal involvement, 30 (29.1%) patients had peripheral arthritis, and 31 individuals (30.1%) had sacroiliac joint involvement. The mean PASI, DAPSA, and DAS28 scores were 10.9 (SD: 4.4), 16.7 (SD: 7.5) and 3.2 (SD: 0.8), respectively.

In our patients cohort, the mean CSI score was 45.4 (SD: 15.1), and a clinically significantly CS (CSI score ≥ 40) was detected in 67 (65.1%) patients. Patients with CS has significantly higher BMIs and disease activity scores than others. Demographic and clinical characteristics of PsA patients with CS versus those without CS are shown in Table 1.

The frequency of CS-related conditions in CSI scores part B was as follows: 6 (5.8%) in restless leg syndrome, 7 (6.7%) in chronic fatigue syndrome, 10 (9.7%) in FM, 6 (5.8%) in migraine or tension headaches, 8 (7.7%) in IBS, 5 (4.8%) in anxiety or panic attacks, and 9 (8.7%) in depression. The mean GAD-7 score of all patients was 7.87 (SD: 4.1), the mean PHQ score was 12.65 (SD: 8.7), the mean ISI score was 13.84 (SD: 6.4), and the mean HAQ score was 10.76 (SD: 4.5). Table 2 provides a summary of additional clinical data.

The number of patients according to CS level was as follows: 16 (15.5%) subclinical, 20 (19.4%) mild, 27 (26.2%) moderate, 25 (24.2%) severe, and 15 (14.5%) excessive. As CS levels worsened, patients tended to have a higher BMI; increased anxiety, depression, sleep quality, and disease activity scores; and diminished functionality. BMI, DAPSA, PASI, DAS28, GAD-7, PHQ-9, ISI, and HAQ scores according to CS levels and the results of post hoc comparisons are presented in Table 3.

Upon correlation analysis, CSI score showed a weak positive correlation with BMI. The severity of psoriasis and disease activity scores of PsA were positively correlated with CSI score (the correlation coefficient ranged from 0.393 to 0.652, *p* < 0.001). Additionally, CSI score was moderately positive correlated with PHQ-9 and HAQ, and strongly positive correlated with GAD-7 and ISI. Table 4 shows the correlation analysis results.

A logistic regression analysis was conducted to assess the impact of specific variables on the occurrence of CS during the final stage of the analysis. The results of the univariate logistic regression analysis indicated a significant association between variable BMI (odds ratio (OR) 1.16; 95% CI: 1.04–1.31) (*p* = 0.009), DAPSA (OR 1.33, 95% CI: 1.18–1.50) (*p* < 0.001), PASI (OR 2.32, 95% CI: 1.69–3.19) (*p* < 0.001), DAS-28 (OR 7.22, 95% CI: 2.24–23.31) (*p* = 0.001), GAD-7 (OR 2,19 95% CI: 1.54–3.10) (*p* < 0.001), PHQ-9 (OR 1.68 95% CI: 1.34–2.10) (*p* < 0.001), ISI (OR 1.42 95% CI: 1.23–1.65) (*p* < 0.001), HAQ (OR 1.92 95% CI: 1.45–2.56) (*p* < 0.001) and the existence of CS. Following this, several multiple regression models were constructed based on the significance values of these variables. The logistic regression analysis revealed that the variables PASI, GAD-7, and ISI exhibit considerable predictive power in relation to the outcome variable CS (*p* < 0.05). PASI was observed as the most important variable in predicting CS (OR 9.70 95% CI: 1.52–62.21). The findings of both univariate and multivariate regression analysis are presented in Table 5.

## 4. Discussion

To improve the effectiveness of treatment and achieve better outcomes in chronic musculoskeletal diseases, it is crucial to identify the type of pain and any accompanying conditions such as anxiety, depression, and sleep disorders. Patients with active inflammatory joint disease may experience neuropathic, nociplastic, or mixed pain in addition to nociceptive pain. In our study of patients with PsA, two out of three patients had clinically significant CS. Additionally, significant associations were found between the CSI score and various factors, including BMI, disease activity scores, levels of anxiety and depression, presence of insomnia, and overall quality of life in our study. The severity of psoriasis, anxiety level, and sleep quality were independent predictors of a worse CSI score.

The etiology of pain mostly stems from dysfunctions within the pain-regulating mechanisms of the CNS. In a manner akin to peripheral sensitization, the dysregulation of pain pathways inside the CNS has the potential to induce hyperalgesia and allodynia [30]. It is plausible that an imbalance between pain-facilitating and pain-inhibiting pathways within the CNS may serve as the underlying mechanism for diseases linked with chronic pain [30]. There are three main groups of pain-regulating systems in the CNS. CS is considered to be the most crucial mechanism for regulating pain in the CNS. CS is a phenomenon that takes place within the dorsal horn of the spinal cord, leading to the enlargement of receptive fields and heightened sensitivity to pain [31]. There are two main phases involved in this process. The first phase is an acute phase, which is facilitated by the activation of N-methyl-D-aspartate (NMDA) receptors by the binding of glutamate. The second phase is a chronic phase, which is characterized by the production of pain-regulating peptides and the activation of spinal microglia [31]. While there have been multiple studies conducted to evaluate CNS pain mechanisms in individuals with FM and research on CNS pain mechanisms in osteoarthritis is growing, the investigation of central pain in rheumatoid arthritis (RA) is still in its early stages. The available psychophysical and neuroimaging evidence indicates that a certain group of osteoarthritis patients exhibit pain sensitization that is mediated by central mechanisms [32]. A study involving a cohort of 282 individuals diagnosed with osteoarthritis revealed a higher prevalence of CS, particularly among those afflicted with hand osteoarthritis [33]. This phenomenon can be attributed to the probable occurrence of supraspinally mediated decreases in inhibitory processes and increases in facilitatory processes of nociceptive communication [32]. The prevalence of CS in RA patients is affected by several factors. These include the symmetrical expression of the disease, the weak correlation between disease activity and symptoms, and widespread hyperalgesia in both articular and non-articular regions in response to various stimuli [34]. In a retrospective observational study including 78 PsA and 72 RA patients, the prevalence of CS was 42.9% and 29%, respectively [15]. The study examined the correlation between CS and Behçet’s disease in patients, revealing a positive presence of CS in 54.5% of the individuals [35]. The presence of CS was detected in 37 out of 50 individuals diagnosed with Sjögren’s syndrome [36]. In the study conducted by Salaffi et al. [37], it was found that disease activity was higher and functional ability and quality of life were worse in PsA patients with CS symptoms. In our study, the rate of CS-positive patients was 65.1%. Zhao et al. [38] systematically analyzed forty scholarly papers to examine the prevalence rates of FM in RA, axial spondyloarthritis (axSpA), and PsA. The findings revealed that the prevalence of FM in RA varied between 4.9% and 52.4%, while in axSpA, it ranged from 4.11% to 25.2%, and in PsA it ranged from 9.6% to 27.2%. The coexistence of FM was found to be correlated with increased disease activity in individuals diagnosed with RA and ankylosing spondylitis (AS). This association was observed by the assessment of disease activity scores, namely the DAS28 for RA and BASDAI for AS [38]. The results revealed that both CS scores and the incidence of FM were significantly greater among Behçet’s patients than in the control group. In our study, although the FM rate was higher in patients with CS than those without CS, it was not statistically significant. Furthermore, it was observed that the prevalence of CS was notably elevated among individuals exhibiting high scores indicating disease activity. Bellinato et al. [39] found that anxiety and depression scores were elevated in PsA patients and exceeded those observed in PsO patients. Nevertheless, a comprehensive assessment could not be conducted due to the absence of a comparison group consisting of individuals without CS. In the present study, it was shown that individuals diagnosed with CS had elevated levels of anxiety and depression compared to those who did not have CS. The findings of the study indicated that persons diagnosed with Behçet’s disease, who had positive results for CS, experienced more severe symptoms of sleeplessness than the control group [35]. It was also observed that patients diagnosed with FMF exhibited lower levels of quality of life, anxiety, depression and insomnia severity in patients with CS compared to patients without CS [40]. In a similar vein, those diagnosed with axSpA had elevated levels of lower quality of life, insomnia severity, depression, and anxiety in comparison to the control group [41]. The observed level of CS in individuals with FMF is directly proportional to the severity of symptoms such as anxiety, sadness, sleeplessness, and overall quality of life, thereby corroborating these findings [40]. Consequently, over an extended period, CS assumes a prominent role in the clinical presentation and exerts detrimental effects on the functionality of a majority of patients across several dimensions.

There were several limitations of this study. Firstly, the study did not gather data on the socio-economic position of the participants, thus precluding any knowledge regarding potential disparities in socio-economic level or health literacy across the groups that could impact the assessment of the questionnaires. Secondly, due to the cross-sectional design of this study, the temporal relationship between disease activity, depression, anxiety, sleep quality, CS and functionality remain unclear. Thirdly, patient selection bias could not be completely eliminated, due to the inclusion of patients admitted to a single center. The fact that PsO patients without PsA were not evaluated in the study can be counted among the limitations of the study.

## 5. Conclusions

The presence of CS was identified in 65% of patients with PsA. Furthermore, a positive correlation was discovered between the degree of CS and various indicators, including disease activity, levels of anxiety and depression, severity of sleeplessness, and diminished quality of life. This study highlights the necessity for a novel viewpoint in treating patients with PsA and understanding the significance of CS in rheumatic conditions.

## Figures and Tables

**Table 1 medicina-60-01449-t001:** Demographic and clinical characteristics of the patients.

Characteristics	CSI Score ≥ 40 (*n* = 67)	CSI Score < 40 (*n* = 36)	*p*
Male, *n* (%)	31 (46.3)	11 (30.6)	0.122
Age, years, mean (SD)	42.6 (5.4)	44.2 (8.6)	0.325
Current smoker, *n* (%)	20 (29.9)	16 (44.4)	0.139
BMI, kg/m^2^, mean (SD)	27.3 (3.3)	25.2 (4.3)	0.014 *
High education level ^a^, *n* (%)	41 (61.2)	15 (41.7)	0.065
Age at diagnosis, years, mean (SD)	38.3 (5.7)	39.5 (7.7)	0.422
Time since diagnosis, years, median (IQR)	4 (2)	4 (3)	0.100
HLA-B27 positivity, *n* (%)	21 (31.3)	10 (27.8)	0.823
Family history of psoriasis, n (%)	25 (37.3)	6 (16.7)	0.014 *
History of IBD, *n* (%)	2 (2.9)	2 (5.6)	0.500
History of uveitis, *n* (%)	16 (23.9)	5 (13.9)	0.382
Entheseal involvement, *n* (%)	41 (61.2)	15 (41.7)	0.058
Peripheral arthritis, *n* (%)	20 (29.9)	10 (27.8)	0.825
Sacroiliac joint involvement, *n* (%)	21 (31.3)	10 (27.8)	0.823
NSAID, *n* (%)	54 (80.5)	26 (72.2)	0.330
csDMARDs, *n* (%)	47 (70.1)	31 (86.1)	0.072
boDMARDs, *n* (%)	21 (31.3)	16 (44.4)	0.186
ESR (mm/h), mean (SD)	21.4 (8.5)	20.8 (10.2)	0.722
CRP (mg/L), mean (SD)	13.1 (4.6)	11.5 (7.4)	0.246
DAPSA, mean (SD)	19.8 (6.4)	10.9 (5.6)	<0.001 *
PASI, mean (SD)	12.9 (3.6)	7.1 (2.7)	<0.001 *
DAS28, mean (SD)	3.4 (0.8)	2.8 (0.6)	<0.001 *

SD—standard deviation; IQR—interquartile range; *—*p* < 0.005; ^a^—defined as international standard classification of education (ISCED) level > 4; IBD—inflammatory bowel disease; BMI—body mass index; NSAID—nonsteroidal anti-inflammatory drugs; boDMARD—biologic originator disease-modifying antirheumatic drugs; csDMARDs—conventional synthetic disease-modifying antirheumatic drugs; DAPSA—Disease Activity Index for Psoriatic Arthritis; PASI—Psoriasis Area Severity Index; DAS28—Disease Activity Score 28; CRP—C-reactive protein; ESR—erythrocyte sedimentation rate.

**Table 2 medicina-60-01449-t002:** Comparison of CSI Part B, anxiety, depression, sleep quality, and quality of life of patients with and without CS.

	CSI Score ≥ 40 (*n* = 67)	CSI Score < 40 (*n* = 36)	*p*
CSI Part B			
1: restless leg syndrome, *n* (%)	5 (7.5)	1 (2.8)	0.333
2: chronic fatigue syndrome, *n* (%)	5 (7.5)	2 (5.6)	0.714
3: fibromyalgia, *n* (%)	9 (13.4)	1 (2.8)	0.082
4: temporomandibular joint disorder, *n* (%)	0 (0.0)	0 (0.0)	
5: migraine or tension headaches, *n* (%)	4 (6.0)	2 (5.6)	0.932
6: irritable bowel syndrome, *n* (%)	6 (9.0)	2 (5.6)	0.539
7: multiple chemical sensitivities, *n* (%)	0 (0.0)	0 (0.0)	
8: neck injury (including whiplash), *n* (%)	0 (0.0)	0 (0.0)	
9: anxiety or panic attacks, *n* (%)	4 (6.0)	1 (2.8)	0.472
10: depression	7 (10.4)	2 (5.6)	0.402
GAD-7, mean (SD)	9.8 (3.5)	4.2 (2.2)	<0.001 *
PHQ-9, mean (SD)	15.1 (5.6)	8.1 (1.8)	<0.001 *
ISI, mean (SD)	16.7 (5.8)	8.4 (3.4)	<0.001 *
HAQ, mean (SD)	12.6 (44)	7.2 (1.4)	<0.001 *

SD—standard deviation; *—*p* < 0.005; ISI—Insomnia Severity Index; GAD-7—General Anxiety Disorder-7; PHQ-9—Patient Health Questionnaire-9; HAQ—health assessment questionnaire; CS—central sensitization; CSI—Central Sensitization Inventory.

**Table 3 medicina-60-01449-t003:** Post hoc results according to patients’ CSI levels.

	Subclinical (*n* = 16) (I)	Mild (*n* = 20) (II)	Moderate (*n* = 27) (III)	Severe (*n* = 25) (IV)	Excessive (*n* = 15) (V)	*p*	Post hoc
BMI, kg/m^2^, mean (SD)	23.10 (4.3)	26.97 (3.6)	27.08 (2.5)	27.29 (3.9)	27.80 (3.5)	0.001 *	I vs. III-IV-V
DAPSA, mean (SD)	5.87 (2.8)	15.00 (3.5)	17.44 (4.1)	20.3 (4.5)	23.1 (10.2)	<0.001 *	I vs. II-III-IV-V, II-V
PASI, mean (SD)	5.26 (1.6)	8.40 (2.6)	11.1 (0.6)	12.9 (1.6)	16.4 (6.1)	<0.001 *	I vs. III-IV-V, II vs. IV-V, V vs. III-IV
DAS28, mean (SD)	2.68 (0.9)	2.90 (0.1)	3.03 (0.1)	3.59 (0.7)	4.09 (1.3)	<0.001 *	I vs. IV-V, V vs. II-III
GAD-7, mean (SD)	2.00 (0.8)	6.00 (0.7)	8.44 (2.7)	10.10 (2.4)	11.87 (5.1)	<0.001 *	I vs. II-III-IV-V, II vs. IV-V, III-V
PHQ-9, mean (SD)	6.62 (1.3)	9.25 (1.1)	12.49 (3.9)	12.80 (3.3)	23.67 (1.3)	<0.001 *	I vs. III-IV-V, II vs. IV-V, V vs. III-IV
ISI, mean (SD)	5.00 (0.8)	11.25 (1.5)	12.19 (1.5)	18.84 (5.8)	21.4 (5.1)	<0.001 *	I vs. II-III-IV-V, II vs. IV-V, III-V
HAQ, mean (SD)	7.87 (1.7)	6.75 (0.8)	11.60 (3.1)	12.40 (3.4)	15.00 (6.8)	<0.001 *	I vs. IV-V, II vs. III-IV-V

SD—standard deviation; *—*p* < 0.005; CSI—Central Sensitization Inventory; DAPSA—disease activity index for psoriatic arthritis; PASI—Psoriasis Area Severity Index; DAS28—disease activity score 28; ISI—Insomnia Severity Index; GAD-7—General Anxiety Disorder-7; PHQ-9—Patient Health Questionnaire-9; HAQ—health assessment questionnaire; BMI—body mass index.

**Table 4 medicina-60-01449-t004:** Correlation between clinical and demographic variables and CSI score.

	Correlation Coefficient	*p*
BMI, kg/m^2^	0.265	0.007 *
DAPSA	0.569	<0.001 *
PASI	0.652	<0.001 *
DAS28	0.393	<0.001 *
GAD-7	0.652	<0.001 *
PHQ-9	0.586	<0.001 *
ISI	0.614	<0.001 *
HAQ	0.578	<0.001 *

*—*p* < 0.005; CSI—Central Sensitization Inventory; DAPSA—Disease Activity Index for Psoriatic Arthritis; PASI—Psoriasis Area Severity Index; DAS28—Disease Activity Score 28; ISI—Insomnia Severity Index; GAD-7—General Anxiety Disorder-7; PHQ-9—Patient Health Questionnaire-9; HAQ—health assessment questionnaire; BMI—body mass index.

**Table 5 medicina-60-01449-t005:** Univariate and multivariate regression analysis examining the relationship between CS and selected parameters and scales.

Covariate	Univariate			Multivariate		
	OR	*p*	95% CI (Lower-Upper)	OR	*p*	95% CI (Lower–Upper)
BMI, kg/m^2^	1.16	0.009	1.04–1.31			
DAPSA	1.33	<0.001	1.18–1.50			
PASI	2.32	<0.001	1.69–3.19	9.70	0.017 *	1.52–62.21
DAS28	7.22	0.001	2.24–23.31			
GAD-7	2.19	<0.001	1.54–3.10	2.89	0.014 *	1.24–6.71
PHQ-9	1.68	<0.001	1.34–2.10			
ISI	1.42	<0.001	1.23–1.65	5.56	0.041 *	1.08–28.72
HAQ	1.92	<0.001	1.45–2.56			

*—*p* < 0.005; CI—confidence interval; OR—odds ratio; DAPSA—Disease Activity Index for Psoriatic Arthritis; PASI—Psoriasis Area Severity Index; DAS28—Disease Activity Score 28; ISI—Insomnia Severity Index; GAD-7—General Anxiety Disorder-7; PHQ-9—Patient Health Questionnaire-9; HAQ—health assessment questionnaire, BMI—body mass index.

## Data Availability

Patient data were collected at Gülhane Training and Research Hospital. The datasets generated and/or analyzed during the current study are available from the corresponding author (MNK) upon reasonable request.

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
