# Peer review of "Impact of Central Sensitization on Clinical and Functional Aspects of Psoriatic Arthritis"

_medicina, 2024, doi:10.3390/medicina60091449_

Round 1

Reviewer 1 Report

Comments and Suggestions for Authors

It is an interesting topic.

However, there are many aspects that need to be improved

Line 96 : ,, body mass index (BMI),

It is an interesting topic.

However, there are many aspects that need to be improved

Line 96 : ,, body mass index (BMI),

Why was this parameter evaluated?

Lines 185-186: ,,As CS levels worsened, patients tended to have higher BMI,…”

How did you evaluate this parameter? How important is BMI in this study?

Line 190: ,,In correlation analysis, CSI score showed a weak positive correlation with BMI”

Line 190: ,,In correlation analysis, CSI score showed a weak positive correlation with BMI”

Can you comment?

Lines 246-247:,,Also, It was found significant associations between the CSI score and various factors, including BMI, disease activity scores…”

Do you have bibliographic references in the specialized literature for these statements?

Lines 3232-326: ,,Furthermore, a positive correlation was discovered between the degree of CS and various indicators, including disease activity, levels of anxiety and depression, severity of sleeplessness, and diminished quality of life.”

Lines 3232-326: ,,Furthermore, a positive correlation was discovered between the degree of CS and various indicators, including disease activity, levels of anxiety and depression, severity of sleeplessness, and diminished quality of life.”

At the conclusions, there is no longer a correlation between BMI and the degree of CS. Can you explain?

Further studies are certainly needed.

I think the conclusions must be improved.

My comments are only intended to make the paper better. Good luck!

Author Response

Dear Reviewer,

Thank you very much for taking the time to review this manuscript.

In the literature, it is thought that there is a positive correlation between the increase in body mass index (BMI) and central sensitization (CS) score, but it has not been fully proven. In our study, we wanted to evaluate the increase in CS score and BMI in psoriatic arthritis patients. Although a correlation was found in the basic analyses, no significant relationship was found in advanced statistical analyses. The articles we used to evaluate the relationship between BMI and CS from the literature are as follows:

1-Kieskamp SC, Paap D, Carbo MJG, Wink F, Bos R, Bootsma H, Arends S, Spoorenberg A. Central sensitization, illness perception and obesity should be considered when interpreting disease activity in axial spondyloarthritis. Rheumatology (Oxford). 2021 Oct 2;60(10):4476-4485.

2- Yücel FN, Gezer HH, Jandaulyet J, Öz N, Acer Kasman S, Duruöz MT. Clinical and functional impact of central sensitization on patients with familial Mediterranean fever: a cross-sectional study. Rheumatol Int. 2023 Jan;43(1):125-136.

3- Sariyildiz A, Coskun Benlidayi I, Turk I, Zengin Acemoglu SS, Unal I. Biopsychosocial factors should be considered when evaluating central sensitization in axial spondyloarthritis. Rheumatol Int. 2023 May;43(5):923-932.

4- Adami G, Gerratana E, Atzeni F, Benini C, Vantaggiato E, Rotta D, Idolazzi L, Rossini M, Gatti D, Fassio A. Is central sensitization an important determinant of functional disability in patients with chronic inflammatory arthritides? Ther Adv Musculoskelet Dis. 2021 Feb 15;13:1759720X21993252.

Sincerely…

Reviewer 2 Report

Comments and Suggestions for Authors

I have gone through manuscript titled "  Impact of Central Sensitization on Clinical and Functional Aspects of Psoriatic Arthritis"  My observations are given below   1. There is no explanation of sample size calculation. 2. Line 167, mean age should be mentioned in units, like 43.2 years 3. it is better to present the statistically significant values with asterisk in super script in all  tables. 4. sub-group analysis based on age and gender could have been done, this information could have  taken from patients. 5. table 3, 2nd column, heading should be subclinical instead of subclinic. 6. language editing is needed. 

Comments on the Quality of English Language

Language needs improvement

Author Response

Dear Reviewer,

Thank you very much for taking the time to review this manuscript.

  1. There is no explanation of sample size calculation.

Comment: In the power analysis, the sample size was taken as effect size 0.5 and was determined as at least 106 patients. The power analysis is attached to the article.

  1. Line 167, mean age should be mentioned in units, like 43.2 years

Comment: In the article, age is expressed as ‘years’ as follows: A hundred three PsA patients with a mean age of 43.2 (SD: 6.7) years

  1. it is better to present the statistically significant values with asterisk in superscript in all tables.

Comment: In all tables, statistically significant values are indicated with an asterisk as superscript. 

  1. sub-group analysis based on age and gender could have been done, this information could have taken from patients.

Comment: First of all, thank you for this very justified criticism, but due to the large number of tables and data, subgroup analyses according to age and gender could not be performed.

  1. table 3, 2nd column, heading should be subclinical instead of subclinic.

Comment: The heading in Table 3, column 2, has been changed to subclinical.

  1. language editing is needed.

Comment: The language of the article was revised by two native English speakers who are experts in medical English.

Sincerely…

Reviewer 3 Report

Comments and Suggestions for Authors

The present study by Nur Kaya and colleagues examines the prevalence of central sensitization and its impact on the clinical and functional aspects of psoriatic arthritis. Observations and suggestions are listed below:

1. The abbreviation for the CASPAR criterion must be described in the abstract.

2. What is the worldwide prevalence of psoriatic arthritis in Turkey?

3. Is there any specific treatment for this disease, or is it like rheumatoid arthritis?

4. Should the phrase in lines 48-50 be enclosed in quotation marks? Only the apostrophe appears.

5. On line 260, it is necessary to describe the abbreviation for NMDA.

6. Could the treatment patients receive impact their pain perception in this analysis?

7. Add the following reference to the discussion: Salaffi F et al. PMID: 37967915 DOI: 10.3899/jrheum.2023-0177

Comments on the Quality of English Language

The article is of high quality at the English level. However, some details of writing and syntax need to be addressed.

Author Response

Dear Reviewer,

Thank you very much for taking the time to review this manuscript.

  1. The abbreviation for the CASPAR criterion must be described in the abstract.

Comment: The abbreviation for the CASPAR criterion is explained in the abstract section.

  1. What is the worldwide prevalence of psoriatic arthritis in Turkey?

Comment: The prevalence of psoriatic arthritis in Turkey is estimated to be 10-14%.

(Alinaghi F, Calov M, Kristensen LE, et al. Prevalence of psoriatic arthritis in patients with psoriasis: A systematic review and meta-analysis of observational and clinical studies. J Am Acad Dermatol. 2019 Jan;80(1):251-265.e19.)

  1. Is there any specific treatment for this disease, or is it like rheumatoid arthritis?

Comment: Psoriatic arthritis (PsA) is an inflammatory musculoskeletal disease associated with psoriasis that was initially considered a variant of rheumatoid arthritis but later emerged as a distinct clinical entity. The term seronegative arthritis is generally most appropriate for PsA. Initial treatments are similar, but differ depending on the organ involved. Treatment is initially guided by an assessment of disease severity, including the degree of disease activity, damage and impact on the patient for each clinical area. A targeted treatment approach should be applied, targeting remission/inactive disease or low/minimal disease activity.

  1. Should the phrase in lines 48-50 be enclosed in quotation marks? Only the apostrophe appears.

Comment: Apostrophe deleted from lines 48-50.

  1. On line 260, it is necessary to describe the abbreviation for NMDA.

Comment: In line 260 the abbreviation for NMDA is explained.

  1. Could the treatment patients receive impact their pain perception in this analysis?

Comment: Medical treatments that could affect patients' pain perception were excluded from the study.

  1. Add the following reference to the discussion: Salaffi F et al. PMID: 37967915 DOI: 10.3899/jrheum.2023-0177

Comment: Added ‘Salaffi F et al. PMID: 37967915 DOI: 10.3899/jrheum.2023-0177’

reference to discussion.

Sincerely…
